# A Fast and scalable Extended Hoyer Projection for Structured Neural Network Sparsity

## Abstract

Deep networks require sparsity mechanisms that are both scale-invariant and computationally efficient. Existing approaches based on the Hoyer score rely on nonconvex projections, resulting in unstable heuristics and potential convergence issues.

In this paper, we introduce a new Cone Alignement Index (CAI), a convex constraint whose level sets form a Lorentz hypercone. This geometric structure enables the first Closed-Form Projection (CFP) onto such a cone, requiring only a single interpolation step and enjoying guaranteed convergence. We derive analytical expressions for: (i) computing the active set through a provably correct threshold rule, and (ii) performing the final projection using a closed-form interpolation coefficient.

Building on this result, we propose a fast bilevel projection method, consisting solely of successive Closed-Form Projection (CFP) algorithms, with guaranteed convergence and naturally inducing hardware-friendly column (or row)-wise sparsity.

Thanks to these Closed-Form Projection (CFP) algorithms, our method is up to 6.5 times faster than the original Hoyer projection on the vector. Our bilevel Closed-Form Projection (CFP) algorithm is 2r times faster than the HALS algorithm on matrices. Applied to transformer attention matrices on biomedical and NLP dataset (GLUE benchmark), it achieves up to $96\%$ sparsity with negligible accuracy degradation, outperforming state-of-the-art "universal Big bird" masks.

Overall, this work provides a principled, convex, and scalable alternative to Hoyer-based sparsification, opening the door to energy-efficient LLMs with controllable structured sparsity.

## 1 State of the art of neural network sparsification

Modern deep neural networks (DNNs) achieve state-of-the-art performance across a wide range of tasks due to their substantial capacity, typically achieved through a huge number of trainable parameters Krizhevsky et al. (2012); He et al. (2016); Vaswani et al. (2017). However, this parameter abundance entails significant computational and memory requirements, which lead to a huge carbon footprint during training and inference. To address these challenges, a large literature has focused on neural network sparsification: the process of reducing the number of non-zero weights in a model. One of the earliest and most widely adopted approaches to induce sparsity in neural networks was the pruning method Alvarez & Salzmann (2016); Han et al. (2015); Frankle & Carbin (2019). Classical pruning methods citeSanh, RigL Evci et al. (2021), and Sparse GPT Frantar & Alistarh (2023) eliminate weights using magnitude-based or gradient-based heuristics. These approaches produce empirical sparsity but without geometric constraints. Advanced structured pruning methods Xia et al. (2024) Ashkboos et al. (2024), overcomes the inefficiency of classical pruning methods.
The Least Absolute Shrinkage and Selection Operator (LASSO) Tibshirani (1996); Hastie et al. (2015) penalize the $\ell_1$-norm. The $\ell_0$ norm, which directly counts the number of non-zero weights, offers perfect sparsity control and is scale-invariant but is non-differentiable Louizos et al. (2018). A key limitation of pruning methods, and $\ell_1$ and $\ell_0$-induced sparsity is its unstructured nature, which tends to produce random zero-valued weights. A lot of modern hardware performs the multiply-add operation in a single instruction. This irregular pattern fails to translate into practical computational

speed-ups on these hardware accelerators and is incompatible with efficient parallel processing.

To overcome the inefficiency of unstructured sparsity, research has turned toward structured sparsity methods, which aim to remove entire groups of parameters such as filters or neurons. Group LASSO and its variants introduce regularizes that enforce sparsity at the group level Yuan & Lin (2006); Kim & Xing (2010); Scardapane et al. (2017); Yoon & Hwang (2017); Simon et al. (2013); Wen et al. (2016); Ma et al. (2019); Alvarez & Salzmann (2016). Despite their improved hardware efficiency, these methods still suffer from the computational overhead associated with solving complex Lagrangian optimization problems Friedman et al. (2010); Mairal & Yu (2012).

An alternative to Lagrangian regularization is optimization under constraints using $\ell_1$ projection methods. These methods directly enforce sparsity by projecting weight vectors onto norm balls, typically the $\ell_1$ norm ball, using efficient algorithms Duchi et al. (2008); Condat (2016); Perez et al. (2019). $\ell_1$ projection-based sparsification benefits from linear-time complexity, but is not scale invariant and does not induce structured sparsity.Of particular interest is the $\ell_{1,\infty}$ projection, which enforces structured sparsity by promoting group-wise shrinkage—e.g., setting entire columns of a weight matrix to zero. Recent work has proposed efficient algorithms for $\ell_{1,\infty}$ projection based on the Moreau proximal identity Moreau (1965); Bauschke & Combettes (2017); Condat et al. (2023) Bejar et al. (2021); Quattoni et al. (2009). However, the worst-case time complexity of these algorithms remains $\mathcal{O}(nm\log(nm))$, which may hinder their scalability to very large neural networks.

A promising alternative is the Hoyer score, introduced in Hoyer (2004), which balances sparsity and scale invariance. It has been successfully applied in contexts such as blind deconvolution Repetti et al. (2015), non-negative least squares Esser et al. (2013)Gillis & Glineur (2012), neural network regularization Yang et al. (2020),Ohib et al. (2022), Thom et al. (2015) and biomedical applications Duan et al. (2019).

Large pretrained Transformer models such as BERT Devlin et al. (2019)and RoBERTa Liu et al. (2020) have defined the modern landscape of NLP. These networks are fully dense and employ a standard self-attention mechanism with quadratic complexity $\mathcal{O}(n^2)$ in sequence length $n$. Structured-sparse attention mechanisms have been explored in BigBird Zaheer et al. (2020b;a), and reformer Kitaev et al. (2020). These methods reduce complexity through architectural biases but do not solve a principled optimization problem.

### 1.1 CONTRIBUTION AND ORGANIZATION OF THIS WORK

In this work, we provide the following contributions: i) A new Cone Alignement Index (CAI) (referred as extended Hoyer score). ii) A Closed-Form Projection (CFP) algorithm with selection of the active set using a threshold which performs a single projection onto the cone (never used in machine learning to the best of our knowledge), iii) An extension to structured sparsity via a bilevel projection, enabling structured column-wise sparsity in neural networks. iv) An empirical benchmark on classification tasks, on Transformer architecture, demonstrating both accuracy performance and significant sparsity.

## 2 MATHEMATICAL PROPERTIES OF THE NEW CONE ALIGNEMENT INDEX (CAI)

### 2.1 A NEW CONE ALIGNEMENT INDEX (CAI)

Let define the *a Cone Alignement Index (CAI)* of a vector $\boldsymbol{x} \in \mathbb{R}^n$ as

$$H_e(\boldsymbol{x}) = \frac{(\sum_{i=1}^n x_i)^2}{\sum_{i=1}^n x_i^2} = \frac{(\mathbf{1}^\top \boldsymbol{x})^2}{\boldsymbol{x}^\top \boldsymbol{x}}, \tag{1}$$

where $\mathbf{1}$ denotes the all-ones vector in $\mathbb{R}^n$.

**Lemma 2.1.** *Geometric structure. The level sets of $H_e(\boldsymbol{x})$ define a family of* second-order surfaces

$$(\mathbf{1}^\top \boldsymbol{x})^2 = l \|\boldsymbol{x}\|_2^2, \tag{2}$$

*which can be rewritten as*

$$\langle u, x \rangle^2 = \frac{l}{n} \|x\|_2^2.$$

*with $u = \mathbf{1}/\sqrt{n}$ the unit vector along the diagonal axis.*

*This equation corresponds to the boundary of a revolution hypercone with apex at the origin and axis along the diagonal direction $\mathbf{1} = (1, 1, \ldots, 1)$ and aperture angle $\delta = \arccos\left(\sqrt{l/n}\right)$. . For $l \in [0, n]$, the quantity $H_e(\boldsymbol{x})$ measures how well $\boldsymbol{x}$ is aligned with this diagonal axis: $H_e(\boldsymbol{x}) = n$ if $\boldsymbol{x}$ is collinear with $\mathbf{1}$, and $H_e(\boldsymbol{x}) = 0$ if $\boldsymbol{x}$ is orthogonal to it. The interior of this cone is convex, while its boundary corresponds to a quadratic (Lorentz-type) cone.*

**Relation with the Hoyer score.**   The Hoyer score $H(\boldsymbol{x})$ was originally defined as the square of the ratio between $\ell_1$ and $\ell_2$ norms of the vector $\boldsymbol{x}$ Hoyer (2004) and update following Yang et al. (2020):

$$H(\boldsymbol{x}) = \left( \frac{|\boldsymbol{x}|_1}{|\boldsymbol{x}|_2} \right)^2 \tag{3}$$

Unlike the original Hoyer ratio $\|\boldsymbol{x}\|_1/\|\boldsymbol{x}\|_2$, which is non-convex, the Cone Alignement Index (CAI) $H_e(\boldsymbol{x})$ leads to convex cone level sets, making it more suitable for optimization and projection-based algorithms.

**Relation with GSP constraint.**   The GSP constraint (Group sparse Projection) following the definition Ohib et al. (2022) is given by :

$$GSP(\boldsymbol{x}) = \left( \sum_{i=1}^{r} \frac{\sqrt{n_i} - |x_i|_1}{\sqrt{n_i} - 1} \right) \tag{4}$$

We emphasize that this GSP constraint is mathematically different from our Cone Alignment Index (CAI) without *second-order (Lorentz) revolution hypercone geometry*.

| Property | Cone Alignment Index (CAI) | Hoyer | GSP |
|---|---|---|---|
| Convex Lorentz Cone geometry | Yes | No | No |
| Ratio norm constraint | No | Yes | Yes |
| Iterative algorithm | No (Closed-Form Projection (CFP) | Yes | Yes |

Table 1: Comparison between Cone Alignment Index (CAI) projection, the Hoyer projection and the GSP projection.

**Lemma 2.2.** $H_e$ *is **scale-invariant**, as a direct consequence of the definition of the CAI.*

This scale-invariance property yields the following lemma:

**Lemma 2.3.** *The projection $\boldsymbol{x}$ of a point $\boldsymbol{y}$ onto $\mathcal{H}_e$ satisfies*

$$\langle \boldsymbol{x} , \boldsymbol{x} \rangle = \langle \boldsymbol{x} , \boldsymbol{y} \rangle \quad \Longleftrightarrow \quad \|\boldsymbol{x}\|_2 = \sqrt{\langle \boldsymbol{x} , \boldsymbol{y} \rangle}. \tag{5}$$

*As a consequence, once the line containing the projection point is known, the optimal norm of $\boldsymbol{x}$ can be directly computed. Substituting this relation into the objective yields*

$$\|\boldsymbol{x} - \boldsymbol{y}\|_2^2 = \|\boldsymbol{y}\|_2^2 - \langle \boldsymbol{x} , \boldsymbol{y} \rangle \tag{6}$$

*which shows that the objective is minimized when $\langle \boldsymbol{x} , \boldsymbol{y} \rangle$ is maximized, i.e., when the angle between $\boldsymbol{x}$ and $\boldsymbol{y}$ is minimized.*

## 3   CONE ALIGNMENT INDEX (CAI) PROJECTION

### 3.1   ITERATIVE HYPERCONE PROJECTION ALGORITHM

Since the Cone Alignment Index (CAI) cone is a convex (Lorentz) cone, projection onto it is relatively straightforward (convex optimization). We adopt the classical interpolation:

$$\boldsymbol{x} = \lambda \boldsymbol{y} + (1 - \lambda)\boldsymbol{d}. \tag{7}$$

Substituting this expression into the Cone Alignment Index (CAI) yields the following quadratic equation in $\lambda$:

$$a\lambda^2 + b\lambda + c = 0 \tag{8}$$

with coefficients

$$
\begin{aligned}
a &= \ell_1^2 - l\ell_2^2 + \rho(n-l)(n\rho - 2\ell_1), \\
b &= 2\rho(n-l)(\ell_1 - n\rho), \\
c &= n\rho^2(n-l),
\end{aligned}
\tag{9}
$$

where $\boldsymbol{d} = (\rho, \rho, \ldots, \rho)$ and $\ell_i$ denotes the $\ell_i$-norm of $\boldsymbol{y}$. This quadratic equation always admits two real solutions, except in the degenerate case, where $\boldsymbol{y}$ lies exactly on the hypercone axis. Proofs are provided in supplementary material (appendix).

**Choosing an Efficient Value for $\boldsymbol{d}$**  Selecting $\boldsymbol{d}$ such that $\|\boldsymbol{d}\|_1 = \|\boldsymbol{y}\|_1$ yields a simplified quadratic equation, since the coefficient $b$ vanishes:

$$
a = l\left(\frac{\ell_1^2}{n} - \ell_2^2\right), \quad b = 0, \quad c = \ell_1^2\left(1 - \frac{l}{n}\right).
\tag{10}
$$

This leads to the following Closed-Form Projection (CFP) solution for $\lambda$:

$$
\boxed{\lambda = \sqrt{\frac{\ell_1^2\left(\frac{l}{n} - 1\right)}{l\left(\frac{\ell_1^2}{n} - \ell_2^2\right)}} = \sqrt{\frac{H(\boldsymbol{y})(n-l)}{l(n - H(\boldsymbol{y}))}}.}
\tag{11}
$$

**Iterative Cone Alignement Index (CAI) Projection**  Based on these lemmas, we propose the following iterative algorithm:Ensure all components of $\boldsymbol{y}$ are nonnegative. i) Compute the projection of $\boldsymbol{y}$ onto $\mathcal{H}_e$. ii) Compute the projection onto $\mathbb{R}_+^n$. iii) Iterate between these two projections until the projection onto $\mathcal{H}_e$ lies inside $\mathbb{R}_+^n$, and therefore belongs to $\mathcal{H}_s$.

Finally, restore the original signs of $\boldsymbol{y}$ and rescale to satisfy relation 5. Following Remark 2.2, the generating line can be obtained by considering two points lying on the diametral hyperplane that contains $\boldsymbol{y}$ and computing their intersection with $\mathcal{H}_e$. In practice, we approximate this step using interpolation (Equation 7) with $\boldsymbol{y}$ and $\boldsymbol{d}$, where $\boldsymbol{d}$ lies on the axis of the revolution hypercone. In our implementation, we enforce $\|\boldsymbol{d}\|_1 = \|\boldsymbol{y}\|_1$.

---

**Algorithm 1** Iterative Hypercone Projection Algorithm

---

**Input:** $\boldsymbol{y}, l$
$\boldsymbol{x}_i \leftarrow |\boldsymbol{y}_i|, \ \forall i \in [1, \ldots, n]$
**while** $H(\boldsymbol{x}) > l$ **do**
  $\nu \leftarrow \ell_0(\boldsymbol{x})$ *(hyperplane dimension)*
  $\boldsymbol{d} \leftarrow \left(\frac{\ell_1}{\nu} \text{ if } \boldsymbol{x}_j \neq 0 \text{ else } 0 \quad \forall j \in [1, \ldots, n]\right)$
  $\lambda \leftarrow \sqrt{\frac{H(\boldsymbol{x})(\nu-l)}{l(\nu - H(\boldsymbol{x}))}}$
  $\boldsymbol{x} \leftarrow \lambda\boldsymbol{x} + (1-\lambda)\boldsymbol{d}$
  $\boldsymbol{x}_i \leftarrow \max(0, \boldsymbol{x}_i), \ \forall i \in [1, \ldots, n]$
**end while**
$\boldsymbol{x}_i \leftarrow \boldsymbol{x}_i \times \text{sign}(\boldsymbol{y}_i), \ \forall i \in [1, \ldots, n]$ *(restore sign)*
$\boldsymbol{x} \leftarrow \boldsymbol{x}\frac{\langle \boldsymbol{x}, \boldsymbol{y}\rangle}{\langle \boldsymbol{x}, \boldsymbol{x}\rangle}$ *(normalize using relation 5)*
**Output:** $\boldsymbol{x}$

---

where $\lambda$ is the interpolation coefficient.

### 3.2 A CLOSED-FORM PROJECTION (CFP) ALGORITHM PERFORMING A SINGLE PROJECTION ONTO THE LORENTZ HYPERCONE

The main drawback with this iterative algorithm is its computational cost and the potential convergence issue. Thus, we propose the following Closed-Form Projection (CFP) algorithm.

**Lemma 3.1.** *Using Equation 7, the following condition holds:*

$$
\boldsymbol{x}_i \geq 0 \quad \Rightarrow \quad \lambda(\boldsymbol{y}_i - \rho) + \rho \geq 0.
\tag{12}
$$

*Then, from Equation 12, we obtain the following threshold: any component $\boldsymbol{y}_i$ smaller than $\alpha$ will be projected to zero.*

$$
\boxed{
\begin{aligned}
\boldsymbol{y}_i &\geq \frac{\lambda-1}{\lambda}\frac{\ell_1}{n} = \alpha \\
\alpha &= n^{-1}\ell_1\left(1 - \sqrt{\frac{l(n-H(\boldsymbol{y}))}{H(\boldsymbol{y})(n-l)}}\right).
\end{aligned}
}
\tag{13}
$$

Thanks to the closed-form of Equation 13, we can identify which components of $\boldsymbol{y}$ must be set to zero without explicitly computing the projection onto the hypercone $\mathcal{H}_e$. Consequently, the projection is required only once, at the final step, since every point lying in the plane generated by the hypercone axis and the point $\boldsymbol{y}$ converges to the same generating line. Based on these lemmas, we propose the following procedure:i) Ensure all components of $\boldsymbol{y}$ are nonnegative. ii) Determine the active set using the closed form threshold (Equation 13). iii) Compute the projection using one interpolation using the closed-form of $\lambda$

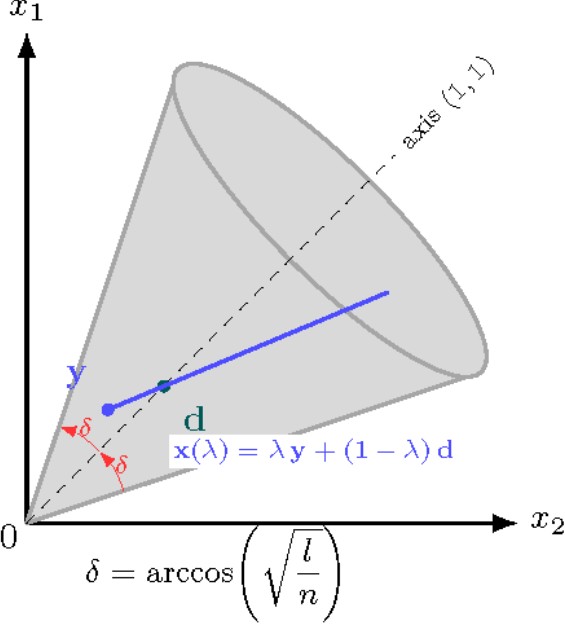

Figure 1: 2D illustration of the interpolation

---

**Algorithm 2** A Closed-Form Projection (CFP) algorithm performing a single projection onto the Lorentz Hypercone

---

**Input:** $\boldsymbol{y}, l$
$\boldsymbol{x}_i \leftarrow |\boldsymbol{y}_i|, \quad \forall i \in [1, \ldots, n]$
$\alpha \leftarrow 0$
$\nu \leftarrow \ell_0(\boldsymbol{x}) + 1$
**while** $\nu \neq \ell_0(\boldsymbol{x})$ **do**
   $\nu \leftarrow \ell_0(\boldsymbol{x})$ *(hyperplane dimension)*
   $\alpha \leftarrow \nu^{-1} \ell_1 \left( 1 - \sqrt{\frac{l(\nu - H(\boldsymbol{x}))}{H(\boldsymbol{x})(\nu - l)}} \right)$
   $\boldsymbol{x} \leftarrow (\boldsymbol{x}_i \text{ if } \boldsymbol{x}_i \geq \alpha \text{ else } 0 \quad \forall i \in [1, \ldots, n])$
**end while**
$\lambda \leftarrow \frac{1}{1 - \frac{\alpha\nu}{\ell_1}}$ *(recompute $\lambda$)*
$\boldsymbol{d} \leftarrow \left( \frac{\ell_1}{\nu} \text{ if } \boldsymbol{x}_j \neq 0 \text{ else } 0 \quad \forall j \in [1, \ldots, n] \right)$
$\boldsymbol{x} \leftarrow \lambda \boldsymbol{x} + (1 - \lambda) \boldsymbol{d}$
$\boldsymbol{x}_i \leftarrow \boldsymbol{x}_i \times \text{sign}(\boldsymbol{y}_i), \quad \forall i \in [1, \ldots, n]$ *(restore signs)*
$\boldsymbol{x} \leftarrow \boldsymbol{x} \frac{\langle \boldsymbol{x}, \boldsymbol{y} \rangle}{\langle \boldsymbol{x}, \boldsymbol{x} \rangle}$ *(normalize using relation 5)*
**Output:** $\boldsymbol{x}$

---

where $\lambda$ is the interpolation coefficient, $\alpha$ is the threshold and $\nu$ is the active set size.

The while loop in this algorithm simply determines the number of components that must be set to zero. The key difference with the iterative algorithm is that the fast algorithm requires only one interpolation step, which guaranteed convergence.

**Theorem 3.2** (Finite-time convergence of the active set selection ). *Given a threshold $\alpha \geq 0$, we define the hard-thresholding operator $T_\alpha : \mathbb{R}_+^n \to \mathbb{R}_+^n$ by*

$$\big(T_\alpha(\boldsymbol{x})\big)_i = \begin{cases} x_i, & \text{if } x_i \geq \alpha, \\ 0, & \text{otherwise}, \end{cases} \qquad i = 1, \ldots, n. \tag{14}$$

*The fixed-point equation*

$$\boldsymbol{x} = T_{\alpha(\boldsymbol{x})}(\boldsymbol{x}) \tag{15}$$

*captures the idea that the support of $\boldsymbol{x}$ and the threshold $\alpha(\boldsymbol{x})$ must be mutually consistent: the entries below the threshold are zeroed out, and the threshold itself is computed from the nonzero entries only. The iterative loop for computing $\alpha$ converges in at most $n$ iterations to a fixed point of equation 15. More precisely, there exists $K \leq n$ such that*

$$\boxed{\begin{aligned} \boldsymbol{x}^{(K+1)} &= \boldsymbol{x}^{(K)}, \\ \boldsymbol{x}^{(K)} &= T_{\alpha(\boldsymbol{x}^{(K)})}\big(\boldsymbol{x}^{(K)}\big). \end{aligned}} \tag{16}$$

### 3.3 BILEVEL CONE ALIGNMENT INDEX (CAI) PROJECTION

Let define the $\ell_\infty$ norm of a vector $\boldsymbol{y}$ is

$$\ell_\infty(\boldsymbol{y}) = \max_{i=1,\ldots,n} y_i \tag{17}$$

The $\ell_{1,\infty}$ projection enforces structured sparsity by promoting group-wise shrinkage, setting entire columns of a weight matrix to zero. This property significantly enhances computational efficiency. However, since the Hoyer score is not a norm, we cannot derive an efficient algorithm for $\ell_{H,\infty}$ projection based on the Moreau proximal identity Moreau (1965); Bauschke & Combettes (2017); Bejar et al. (2021). In this paper, we propose an alternative based on a bilevel method Zhang et al. (2022; 2024b); Barlaud et al. (2024). Specifically, we propose a bilevel $\ell_{H,\infty}.projection$. Let $\boldsymbol{Y}$ be a matrix with $n$ rows and $m$ columns, and let $\boldsymbol{y}_1, \ldots, \boldsymbol{y}_n$ denote its column vectors. Let define the row vector composed of the infinity norms of the columns of $\boldsymbol{Y}$.

$$\boldsymbol{v}_\infty = (\|\boldsymbol{y}_1\|_\infty, \ldots, \|\boldsymbol{y}_n\|_\infty), \tag{18}$$

The bilevel projection optimization problem is then defined as:

$$BP_l^{H,\infty}(\boldsymbol{Y}) = \{\boldsymbol{x} \mid \forall j,$$
$$\boldsymbol{x}_j = \arg\min_{\boldsymbol{x}} \|\boldsymbol{x} - \boldsymbol{y}_j\|_2 \quad \text{s.t. } P^\infty(\boldsymbol{x}_j) < u_j\},$$
$$\text{where } \hat{u} \in \arg\min_u \|u - \boldsymbol{v}_\infty\|_2 \quad \text{s.t. } P^H(u) < l. \tag{19}$$

A possible implementation is provided below:

---

**Algorithm 3** Bilevel $\ell_{H,\infty}$ Projection $(BP_\eta^{H,\infty}(\boldsymbol{Y}))$

---

**Input:** $\boldsymbol{Y}, \eta$
$u \leftarrow P_l^H(\|\boldsymbol{y}_1\|_\infty, \ldots, \|\boldsymbol{y}_n\|_\infty)$
**for** $j \in [1, \ldots, n]$ **do**
$\quad \boldsymbol{x}_j \leftarrow P_{u_j}^\infty(\boldsymbol{y}_j)$
**end for**
**Output:** $\boldsymbol{x}$

---

Note that the Closed-form projection and the $\ell_\infty$ projection are closed-form algorithms, which guaranteed convergence of the bilevel algorithm.

## 4 EXPERIMENTAL RESULTS

### 4.1 BENCHMARK OF THE FAST CLOSED-FORM PROJECTION

For the implementation of the original iterative Hoyer projection, we use the efficient projection onto the $\ell_1$ ball initially proposed in Duchi et al. (2008) and later corrected in Condat (2016). Although the empirical computational cost of this projection is $\mathcal{O}(m)$, no theoretical proof of this complexity currently exists.

We use the torch.Profiler which counts operations at the PyTorch level, not at the hardware level. It tracks the computational graph and sums up flops based on the operations executed in the forward pass. As long as the code and inputs are identical, the count should be consistent across devices. Our code (available in supplementary material) reports the same number of flops (floating-point operations) across different GPUs such as NVIDIA or CPU such as Apple M3 or Intel, assuming the same input data, algorithm, and PyTorch version. Based on this metric, Figure 2 shows that the Closed-Form Projection (CFP) fast algorithm has a complexity $KmwithK \approx 6$ and is approximately 6.5 times faster than the original Hoyer projection (depending slightly on the data distribution).

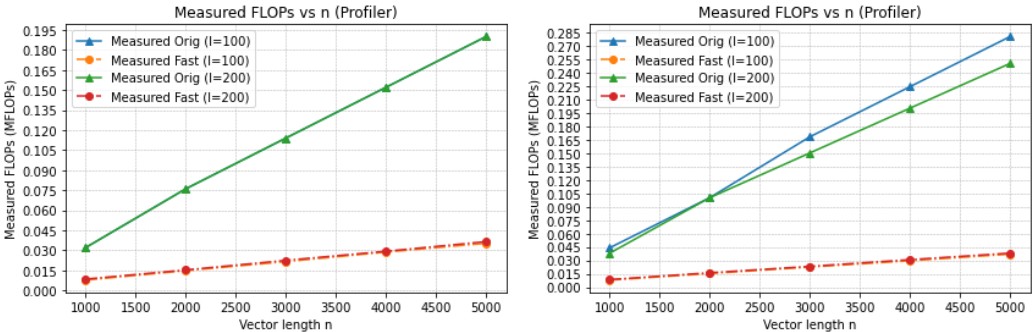

Figure 2: Comparison of two algorithms CFP versus original: Flops Left Gaussian; Right Uniform

For an $m \times m$ attention matrix, the complexity of the bilevel algorithm is K*m for the projection + m*(m-1) for calculating the norm of each column (or row) and 1 flop for the clamp for each column. Therefore, the total flops = $m(m-1) + Km + m$ (with K=6), or approximately $\approx m^2$ flops.

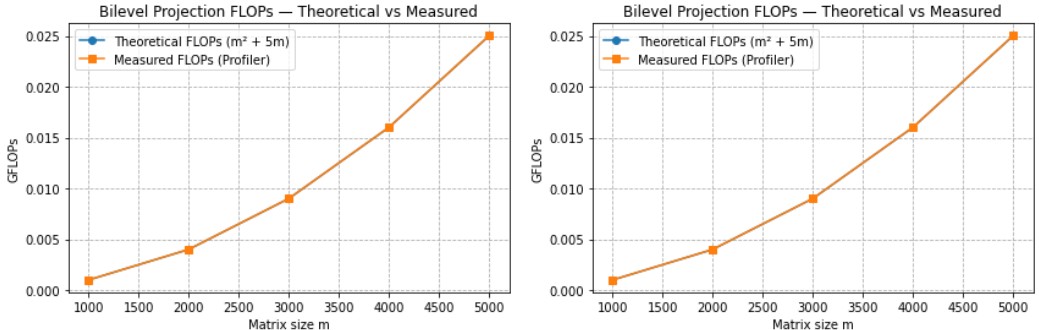

Figure 3: computational cost of the bilevel algorithm: Flops Left Gaussian; Right Uniform

The figure 3shows the perfect match between theoretical and measured flops of our bilevel algorithm. The computational cost for the HALS algorithm Gillis & Glineur (2012) for an $mxm$ attention matrix is total flops = $r(2m^2 + 4mr + m)$ flops, where r is the rank of the matrix Gillis & Glineur (2012), thus approximately $\approx 2rm^2$ flops. Therefore, our bilevel algorithm is $2r$ times faster than the HALS algorithm. A-HALS is faster than HALS in practice, but since even the first iteration of A-HALS (which is the same as the first iteration of HALS) is already more expensive than our bilevel projection (even with r=1 or r=2), A-HALS remains less efficient than our CFP.

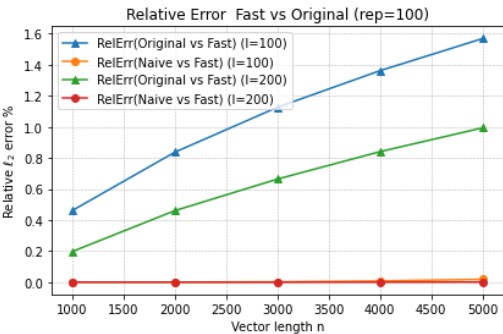

Figure 4: Comparison of two algorithms CFP versus original: Relative norm comparison of the solutions

As illustrated in Figure 4, the relative norm comparison of the solution of our algorithm versus the original shows that solutions are slightly different (The original constraint is a ratio of norms while it is not for the Cone Alignment Index (CAI) constraint).

### 4.2 Constraint Optimization of attention matrices in Transformers

Let $\boldsymbol{W} \in \mathbb{R}^{m \times m}$ denote the attention matrix, where $m$ is the number of tokens. Let $\boldsymbol{z} \in \mathbb{R}^{m \times 1}$ represent the true labels, and $\boldsymbol{z}^*$ the estimated labels obtained from a soft max classifier. To sparsify the weights $\boldsymbol{W}$ of the attention matrix, we employ the bilevel projection method $BP^{H,\infty}$ as a constraint to enforce structured sparsity in the model. The global optimization criterion is defined as:

$$\underset{\boldsymbol{W}}{\text{minimize}} \quad \phi(\boldsymbol{z}, \boldsymbol{z}^*) \quad \text{subject to} \quad BP^{H,\infty}(\boldsymbol{W}) \le l, \tag{20}$$

where $\phi(\boldsymbol{z}, \boldsymbol{z}^*)$ is the cross-entropy loss.

For minimizing this criterion, we follow the work developed by Frankle & Carbin (2019) who proposed a double descent masked gradient algorithm, as follows: after training a network, set all weights smaller than some threshold to zero, rewind the rest of the weights to their initial configuration, and then retrain the network from this starting configuration but keeping the zero weights frozen (untrained). We replace the thresholding by our bilevel projection.

### 4.3 SPARSIFICATION OF ATTENTION MATRICES IN TRANSFORMER ARCHITECTURES

We implemented our classification method using the PyTorch framework for the model, optimizer, schedulers and loss functions. We chose the ADAM optimizer Kingma & Ba (2015), as the standard optimizer in PyTorch. We use the smooth SiLU activation function.

Generative Pre-trained Transformers (GPT) are a class of large language models (LLMs) that have recently attracted significant attention due to their ability to perform a wide range of natural language processing tasks. However, transformer architectures entail substantial computational costs and carbon footprints Strubell et al. (2019); Faiz et al. (2024). This motivates the exploration of sparsity as a strategy to design more efficient models. In this context, we apply our Fast Extended Hoyer projection to the sparsification of attention matrices in transformer architectures Vaswani et al. (2017), with the aim of reducing computational cost. Specifically, we compare our learned diagonal mask, obtained via bilevel projection, against the uniform diagonal mask of Big bird Zaheer et al. (2020b;a) which performs consistently well overall Tay et al. (2021).

The classification framework is implemented in PyTorch, including the model, optimizer, schedulers, and loss functions. For all sparsity levels and both datasets, we set the number of training epochs to 15, the batch size to 32 and the learning rate to $2 \times 10^{-5}$.

#### 4.3.1 EXPERIMENT ON A BIOMEDICAL DATASET: ECG

There are now requirements for classification and interpretation in biomedical applications, such as Single-cell Chen et al. (2023) and ECG for diagnosis of Heart failure, which is a syndrome with complex clinical manifestations Wagner et al. (2020). In this paper, we report results on the Physio Net ECG dataset Goldberger et al. (2000). The challenge of the PTB Diagnostic ECG Database is formulated into a binary classification task with 10,505 abnormal and 4,045 normal ECG. The signals correspond to electrocardiogram (ECG) shapes of heartbeats for the normal case and the abnormal cases affected by different arrhythmias and myocardial infarction. These signals are preprocessed and segmented, with each segment corresponding to a heartbeat with 187 features.

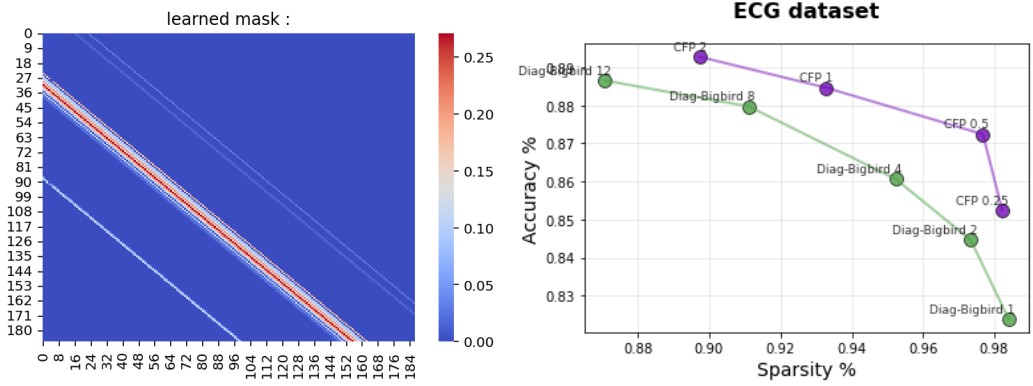

Figure 5: ECG dataset. BigBird versus bilevel $\ell_{H,\infty}$: sparsity–accuracy trade-off.

|  | Baseline | Diagonal BigBird | Diagonal $\ell_{1,\infty}$ |
|---|---|---|---|
| Sparsity (%) | 0 | 97.34 | 97.11 |
| Accuracy (%) | 89.44 | 84.46 | 87.04 |

Table 2: ECG dataset. Comparison of Big bird, and bilevel $\ell_{H,\infty}$: sparsity–accuracy trade-off.

Figure 5 (Left) shows the learned mask obtained with our method; (Right) illustrates that the accuracy curve as a function of sparsity. Our bilevel $\ell_{H,\infty}$ projection outperforms the diagonal Big bird method. As shown in Table 2, for the same sparsity (97%), our learned mask with the bilevel method outperforms the diagonal Bigbird by 3% in accuracy.

### 4.3.2 EXPERIMENT ON A NATURAL LANGUAGE PROCESSING (NLP) TASK

Specifically, we apply our CFP projection to a pretrained transformer-based model Devlin et al. (2019).

We report the accuracy–sparsity trade-off on the GLUE benchmark, focusing on the single-sentence classification task SST-2 Socher et al. (2013). The SST-2 dataset contains approximately 67,000 samples.

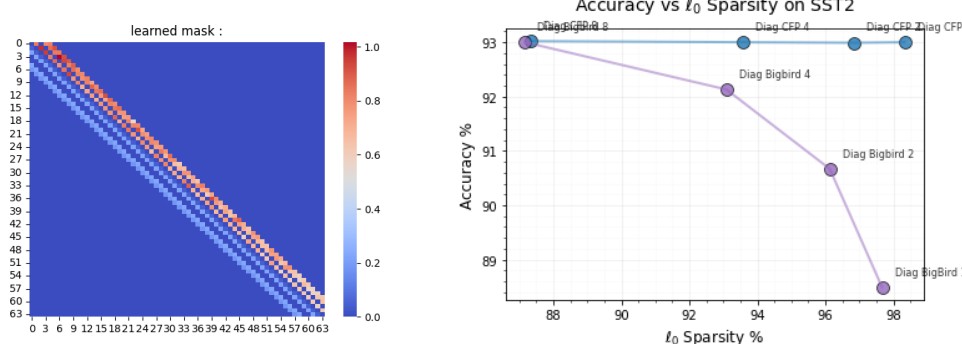

Figure 6: SST-2 dataset. Left: Learned mask using our method. Right: Bigbird versus bilevel $\ell_{1,\infty}$ and $\ell_{H,\infty}$: sparsity–accuracy trade-off.

Figure 6 (Left) shows the learned mask obtained with our method on the second layer of BERT model. Figure 6 (Right) illustrates that the accuracy curve as a function of sparsity is very flat for the CFP projection, while it decreases rapidly for Diagonal Big bird.

|  | Baseline | Diagonal Bigbird 2 | MGPP | Diagonal $\ell_{H,\infty}$ |
|---|---|---|---|---|
| Sparsity (%) | 0 | 92.33 | 90 | 96.11 |
| Accuracy (%) | 92.7 | 91.13 | 90.3 | 92.5 |

Table 3: SST-2 dataset. Comparison of methods for the BERT model: Bigbird, MGPP, and bilevel Hoyer: sparsity–accuracy trade-off.

Table 3 shows that our learned mask with the bilevel method outperforms the diagonal Bigbird mask method by achieving 30% higher sparsity. Our method achieves 96% sparsity with negligible performance degradation of the baseline (full attention matrix). For comparison, we include in table 3 the best results reported in Zhang et al. (2024a).

## 5 DISCUSSION AND CONCLUSION

While pretrained models such as BERT and RoBERTa are fully dense Transformers, some later architectures (e.g., BigBird, Longformer, Reformer) introduce sparse attention mechanisms. However, these models rely on predefined structural masks or heuristic approximations rather than mathematically-grounded sparsity.

In contrast, our method introduces a new Cone Alignment Index (CAI), a convex constraint whose level sets form a Lorentz hypercone.

and the first closed-form projection algorithm requiring a single interpolation operation, with guaranteed convergence and linear complexity. In contrast, our method performs a principled, convex, bilevel projection that analytically determines the active attention support, yielding sparse Transformers with full interpretability and guaranteed convergence.

Our method achieves up to 96% attention sparsity with negligible accuracy loss NLP glue dataset and outperforming state of the art "universal" diagonal Big Bird masks.

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

## A  APPENDIX

### A.1  ANALYSIS OF THE QUADRATIC EQUATION

We consider the following parametric line:

$$\lambda \boldsymbol{y} + (1 - \lambda)\boldsymbol{d},$$

with $\boldsymbol{d} = (\rho, \rho, \ldots, \rho)$, and the extended Hoyer surface $\mathcal{H}_{e,l}$ defined by

$$H_e(\boldsymbol{x}) = l.$$

Solving for the intersection, we obtain:

$$H_e(\lambda \boldsymbol{y} + (1 - \lambda)\boldsymbol{d}) = l$$
$$\Leftrightarrow \frac{\left(\sum_i \lambda(y_i - \rho) + \rho\right)^2}{\sum_i \left(\lambda(y_i - \rho) + \rho\right)^2} = l \tag{21}$$
$$\Leftrightarrow \frac{\lambda^2(\ell_1 - n\rho)^2 + n^2\rho^2 + 2\lambda n\rho(\ell_1 - n\rho)}{\lambda^2(\ell_2^2 + n\rho^2 - 2\rho\ell_1) + n\rho^2 + 2\rho\lambda(\ell_1 - n\rho)} = l.$$

After simplification, this leads to the following quadratic equation in $\lambda$:

$$a\lambda^2 + b\lambda + c = 0, \tag{22}$$

where the coefficients are given by:

$$\boxed{\begin{aligned} a &= \ell_1^2 - l\,\ell_2^2 + (n-l)\left(n\rho^2 - 2\rho\,\ell_1\right), \\ b &= 2(n-l)\rho\left(\ell_1 - n\rho\right), \\ c &= (n-l)n\rho^2. \end{aligned}} \tag{23}$$

Note that when $\ell_1 = n\rho$, i.e., when $\boldsymbol{y}$ lies exactly on the cone axis, the linear term $b$ vanishes and the quadratic reduces to a simpler form.

**Condition on $\boldsymbol{d}$**  using this parameter $\lambda$ with the points $\boldsymbol{y}$ and $\boldsymbol{d}$ provides the following condition for ensuring a positive solution:

$$\rho > n^{-1}\left(\ell_1 - \sqrt{\frac{l(n\ell_2^2 - l\ell_1^2)}{n-l}}\right)$$
$$\Leftrightarrow \quad \|\boldsymbol{d}\|_1 > \|\boldsymbol{y}\|_1 - \sqrt{\frac{l(n\|\boldsymbol{y}\|_2^2 - l\|\boldsymbol{y}\|_1^2)}{n-l}}. \tag{24}$$

**Special case $b = 0$ (choosing $\|d\|_1 = \|y\|_1$).**  If $\ell_1 = n\rho$ (i.e., $b = 0$), the quadratic reduces to $a\lambda^2 + c = 0$ with

$$a = \ell_1^2 - l\,\ell_2^2, \qquad c = n\rho^2(n-l) \;=\; \frac{\ell_1^2}{n}(n-l).$$

Solving for $\lambda > 0$ gives

$$\boxed{\lambda^\star = \sqrt{\frac{c}{-a}} = \sqrt{\frac{\frac{\ell_1^2}{n}(n-l)}{l\,\ell_2^2 - \ell_1^2}} = \sqrt{\frac{H(y)\,(n-l)}{l\,(n - H(y))}}},$$

where $H(y) = \left(\|y\|_1 / \|y\|_2\right)^2$. This is the closed form used in the one-shot projection when $\|d\|_1 = \|y\|_1$.

**Feasibility check.**  After computing $\lambda^\star$, form $x(\lambda^\star) = \lambda^\star y + (1 - \lambda^\star)d$ and project to the correct orthant/sign if needed, then rescale using the relation $\|x\|_2 = \sqrt{x \cdot y}$ to satisfy the projection optimality condition.

## A.2 CONVERGENCE OF THE FAST THRESHOLDING ALGORITHM

Let $\boldsymbol{y} \in \mathbb{R}_+^n$ be a given nonnegative vector (e.g., $|\boldsymbol{y}|$ in our algorithm), and let $\boldsymbol{x} \in \mathbb{R}_+^n$ be a candidate solution. We denote by

$$\nu(\boldsymbol{x}) \;=\; \ell_0(\boldsymbol{x}) = |\{i : x_i \neq 0\}| \quad \text{and} \quad \ell_1(\boldsymbol{x}) = \|\boldsymbol{x}\|_1.$$

We also denote by $H(\boldsymbol{x})$ a sparsity score depending only on the nonzero components of $\boldsymbol{x}$ (e.g., the Hoyer or Cone Alignement Index (CAI)). For a given level $l$ and integer $\nu$, we define the threshold

$$\alpha(\boldsymbol{x}) \;=\; \frac{1}{\nu(\boldsymbol{x})} \, \ell_1(\boldsymbol{x}) \left( 1 - \sqrt{\frac{l\big(\nu(\boldsymbol{x}) - H(\boldsymbol{x})\big)}{H(\boldsymbol{x})\big(\nu(\boldsymbol{x}) - l\big)}} \right), \tag{25}$$

whenever the expression is well-defined. Given a threshold $\alpha \geq 0$, we define the hard-thresholding operator $T_\alpha : \mathbb{R}_+^n \to \mathbb{R}_+^n$ by

$$\big(T_\alpha(\boldsymbol{x})\big)_i = \begin{cases} x_i, & \text{if } x_i \geq \alpha, \\ 0, & \text{otherwise,} \end{cases} \qquad i = 1, \ldots, n. \tag{26}$$

The fixed-point equation

$$\boldsymbol{x} = T_{\alpha(\boldsymbol{x})}(\boldsymbol{x}) \tag{27}$$

captures the idea that the support of $\boldsymbol{x}$ and the threshold $\alpha(\boldsymbol{x})$ must be mutually consistent: the entries below the threshold are zeroed out, and the threshold itself is computed from the nonzero entries only.

We now consider the iterative thresholding scheme used in our fast algorithm. Starting from $\boldsymbol{x}^{(0)} = |\boldsymbol{y}|$, we define the sequence

$$\nu^{(k)} = \ell_0\big(\boldsymbol{x}^{(k)}\big), \tag{28}$$

$$\alpha^{(k)} = \frac{1}{\nu^{(k)}} \, \ell_1\big(\boldsymbol{x}^{(k)}\big) \left( 1 - \sqrt{\frac{l\big(\nu^{(k)} - H(\boldsymbol{x}^{(k)})\big)}{H(\boldsymbol{x}^{(k)})\big(\nu^{(k)} - l\big)}} \right), \tag{29}$$

$$\boldsymbol{x}^{(k+1)} = T_{\alpha^{(k)}}\big(\boldsymbol{x}^{(k)}\big), \tag{30}$$

and stop as soon as the support stabilizes, i.e.,

$$\ell_0\big(\boldsymbol{x}^{(k+1)}\big) = \ell_0\big(\boldsymbol{x}^{(k)}\big).$$

**Lemma A.1** (Monotone support decrease). *For the sequence defined in equation 28, the support sizes satisfy*

$$\nu^{(k+1)} \;\leq\; \nu^{(k)} \quad \text{for all } k,$$

*and $\nu^{(k+1)} < \nu^{(k)}$ whenever $\boldsymbol{x}^{(k+1)} \neq \boldsymbol{x}^{(k)}$.*

*Proof.* By definition of $T_{\alpha^{(k)}}$, the transition from $\boldsymbol{x}^{(k)}$ to $\boldsymbol{x}^{(k+1)}$ can only set some coordinates of $\boldsymbol{x}^{(k)}$ to zero; it never activates new coordinates. Therefore, the number of nonzero entries cannot increase, i.e., $\nu^{(k+1)} \leq \nu^{(k)}$. Moreover, if $\boldsymbol{x}^{(k+1)} \neq \boldsymbol{x}^{(k)}$, at least one coordinate that was previously nonzero is set to zero, hence $\nu^{(k+1)} < \nu^{(k)}$. $\qquad\square$

**Theorem A.2** (Finite-time convergence). *The iterative scheme equation 28 converges in at most $n$ iterations to a fixed point of equation 27. More precisely, there exists $K \leq n$ such that*

$$\boldsymbol{x}^{(K+1)} = \boldsymbol{x}^{(K)},$$

*and $\boldsymbol{x}^{(K)}$ satisfies $\boldsymbol{x}^{(K)} = T_{\alpha(\boldsymbol{x}^{(K)})}\big(\boldsymbol{x}^{(K)}\big).$*

*Proof.* By Lemma A.1, the sequence $\{\nu^{(k)}\}$ is nonincreasing and takes values in $\{0, 1, \ldots, n\}$. Therefore, it must stabilize in at most $n$ steps: there exists $K \leq n$ such that,

$$\nu^{(K+1)} = \nu^{(K)}.$$

By definition of $\boldsymbol{x}^{(K+1)}$, we have $\boldsymbol{x}^{(K+1)} = T_{\alpha^{(K)}}(\boldsymbol{x}^{(K)})$. If the support size is unchanged, then no new zero has been introduced, hence the thresholding operator leaves all nonzero coordinates unchanged. Consequently $\boldsymbol{x}^{(K+1)} = \boldsymbol{x}^{(K)}$, and $\boldsymbol{x}^{(K)}$ is a fixed point of the map $\boldsymbol{x} \mapsto T_{\alpha(\boldsymbol{x})}(\boldsymbol{x})$, which is exactly equation 27. $\qquad\square$

## A.3 BACKGROUND: DENSE VS. SPARSE PRETRAINED TRANSFORMERS

Large pretrained Transformer models such as BERT Devlin et al. (2019) and RoBERTa Liu et al. (2020) have defined the modern landscape of NLP. These networks are fully dense and employ a standard self-attention mechanism with quadratic complexity $\mathcal{O}(n^2)$ in sequence length $n$. However, growing model sizes, energy costs, and the environmental impact of training—including the carbon cost of operating LLMs—motivate the development of sparse alternatives that maintain accuracy while lowering resource consumption. BERT Devlin et al. (2019) introduced bidirectional Transformer pretraining using masked language modeling (MLM) and next-sentence prediction (NSP). RoBERTa Liu et al. (2020) retains the same architecture but modifies the training pipeline: i) removes NSP ii) trains on $10\times$ more data and larger batch sizes; iii) applies dynamic masking. This yields a consistent boost in accuracy across major language benchmarks.

While pretrained models such as BERT and RoBERTa are fully dense, later architectures (e.g., BigBird, Long-former, Reformer) incorporate sparsity through predefined local or random attention masks. However, these methods rely on heuristic or architectural sparsity rather than mathematically grounded constraints. Our work provides the first convex, closed-form projection onto a Lorentz cone, enabling principled sparsification with theoretical guarantees, explicit sparsity control, and interpretability of the resulting attention patterns.

