# OpenReview forum: "A Fast and Scalable Extented Hoyer Projection for Structured Neural Network Sparsity"
_ICLR.cc/2026/Conference — ICLR 2026 Conference Withdrawn Submission_

### Official Review · Reviewer_fkhc · 2025-10-28

**Soundness:** 3
**Presentation:** 3
**Contribution:** 3
**Rating:** 4
**Confidence:** 4

**Summary:**

This paper resolves the non-convex optimization issue associated with achieving structural sparsity with Hoyer-based regularizations. The paper makes theoreitical analysis on the space of Hoyer score and proposes a fast projection algorithm with a single hypercone projection onto the Hoyer score surface. The proposed Hoyer projection leads to improved efficiency-performance tradeoff on autoencoder and transformer tasks.

**Strengths:**

1. This paper makes novel observation and analysis on the surface of Hoyer score. This leads to the first projection-based optimization method with the Hoyer measurement, which is a significant contribution
2. The paper makes solid theoreitical derivations to show the correctness and effectiveness of the proposed projection.
3. The overall presentation is good. The paper is easy to follow and proposed method is well-motivated.

**Weaknesses:**

The major weakness of this work comes with the emperical evaluations. Though the paper claims to resolve the non-convex optimization issue of the ogirinal Hoyer regualrization, whether this is a practically meaningful in deep learning remains doubtful. Deep learning itself is known to be non-convex in most cases, where having a non-convex regualrization added to the loss may not bring much addiitonal difficulty to the optimization. The paper should make a thorough comparison between the Hoyer regularization optimization and the proposed Hoyer projection to show if the proposed method is more effective and/or easier to optimize.

Furthermore, there lacks adequate comparison against related work on sparisity-inducing regualrizations, projections, and neural network pruning methods. The only baseline used in the experiments is L1-based method, which is very basic and has shown to be performing worse than more recent projection and regualrization methods. Adding more baselines, and potentially adding more neural network models into the experiment setting can better justify the method emerically.

**Questions:**

1. Please do a direct coparison of the cost and performance between the proposed Hoyer projection and directly performing gradient descent with Hoyer regularization
2. Please add more baselines, especially traditional projection-based sparisification methods like SCAD and MDP

---

> ### Author Response · Authors · 2025-11-17
> **Cost comparison**
>
> The authors thank the reviewer for his constructive comments.
>
> - Yes in the new version we provide direct comparison of the cost and performance using Flops between the proposed closed-form projection onto the Lorentz hypercone and the Hoyer projection.
>
> - We provide a Pytorch code estimating flops on both vectors and matrices
>
> - We provide a  new convex Cone Alignement Index (CAI)
>
>
> - Novelty 2. A closed-form projection onto the Lorentz hypercone (original contribution).}
> To the best of our knowledge, the algorithm we introduce a closed-form projection onto the Lorentz hypercone using a single interpolation step—is entirely new.
> We derive analytical expressions both for:\\\\
> - the threshold $\alpha$ that determines the active set, and  \\
> We derive analytical expressions both for:\\
> - the threshold $\alpha$ that determines the active set,
> - the interpolation coefficient $\lambda$ that performs the projection.
>
>
> The resulting CFP method proceeds as follows:
> i) Ensure all components of $y$ are nonnegative.
>     ii) Identify the active set using the closed-form threshold $\alpha$.
>     iii) Compute the projection using a single interpolation with the closed-form $\lambda$.
>
> - Scope of applications.
> Our initial submission showed applications across a broad range of domains (autoencoders, single-cell analysis, Otto product classification, etc.).
> Following the reviewers’ comments, we agree to focus the experimentation section on NLP.
> The proposed projection operates equally on a glue dataset.
>
> If granted one additional page, we will include results on a complete and standardized NLP benchmark suite.

---

### Official Review · Reviewer_KKLj · 2025-10-31

**Soundness:** 2
**Presentation:** 2
**Contribution:** 1
**Rating:** 2
**Confidence:** 4

**Summary:**

This paper introduces a projection method for neural network sparsity based on an "extended Hoyer score." The authors describe the geometric properties of this score, showing that its level sets form a hypercone, and leverage this to derive a "one-shot," linear-time projection algorithm. This single-vector method is further extended to a bi-level $\ell_{H,\infty}$ projection to induce structured, column-wise sparsity. The approach is evaluated on autoencoder and transformer architectures, where the authors report improvements in the accuracy–sparsity trade-off.

**Strengths:**

Clear Motivation and Problem Framing: The paper does a good job motivating the need for a scale-invariant, differentiable sparsity-inducing method.

**Weaknesses:**

**Weakness:**

**1. Fundamental lack of novelty leading to re-solving previously solved problems:**
    The major weakness of this paper is a fundamental lack of novelty. Both the Hoyer score and sparsity criterion have been well-studied [1, 2, 3, 4] and in better depth. Specifically, [2, 3] (following from [1] and Hoyer's original work [5]) solve the exact problems this paper addresses  (single-vector and grouped projection, respectively) with far more rigor, theoretical depth, and experimental coverage. The failure to cite or compare against this seminal works fundamentally fails to represent state of the problem the paper is trying to solve.

**2. Algorithms are unproven heuristics with significant theoretical gaps:**
    The paper’s algorithmic framing, based on an "extended Hoyer" score and "hypercone geometry," is a reframing of the standard orthant-projection trick already used in prior work [3]. The algorithms derived from this framing lack theoretical justification: the "naive" alternating projection (Algorithm 1) has no convergence guarantee, and the "fast" algorithms are unproven heuristics for problems that have already been solved *exactly* with rigor.

**2.1 For the single-vector projection (Algorithm 2):**
The paper proposes an iterative hard-thresholding/active-set heuristic without proofs of uniqueness or global optimality. The claimed $O(n)$ complexity is also unsupported by a formal bound on the *while* loop's iterations. Closely related subproblems were already handled by a well-established line of work the authors do not cite: *Potluru et al.* (2013) [1] provided an exact $O(n\log n)$ routine ("Sparse-opt") for the fixed $\ell_1/\ell_2$ constraint, and *Thom et al.* (2015) [2] subsequently developed a provably exact $O(n)$ Euclidean projector via a soft-thresholding Representation Theorem.

**2.2 For the structured-sparsity projection (Algorithm 3):** The proposed bi-level $l_{H,\infty}$ extension (Algorithm 3) is presented as a novel method for structured sparsity. However, it appears to be a simple heuristic composition of a single-vector Hoyer projection and an $l_{\infty}$ projection. The paper provides no proof of feasibility or equivalence to a well-defined constrained optimization problem and fails to compare against rigorous, jointly-optimized group Hoyer projectors like GSP [3].

**Weak Empirical Validation:** The experiments are insufficient to support the paper's claims. The datasets used are either very small (like the 779-sample HIF2 dataset) or are not the most representative benchmarks for Transformer sparsification (like the tabular Otto dataset). They are small-scale, lack statistical rigor (e.g., confidence intervals), and most importantly, omit a direct runtime and accuracy comparison to the most relevant baselines [2, 3]. The reported runtime improvements over the old Hoyer (2004) [5] baseline are modest, and the accuracy differences are not shown to be statistically significant.

Typo/error:
1. Main title on openreview says "extented", do the authors mean extended?
2. Pdf title makes the typo: "EXTENDEDED",  instead of "Extended" again?

**References:**
1. Potluru, Vamsi K., Sergey M. Plis, Jonathan Le Roux, Barak A. Pearlmutter, Vince D. Calhoun, and Thomas P. Hayes. "Block coordinate descent for sparse NMF." _arXiv preprint arXiv:1301.3527_ (2013).
2. Thom, Markus, Matthias Rapp, and Günther Palm. "Efficient dictionary learning with sparseness-enforcing projections." _International Journal of Computer Vision_ 114, no. 2 (2015): 168-194.
3. Ohib, Riyasat, Nicolas Gillis, Niccolo Dalmasso, Sameena Shah, Vamsi K. Potluru, and Sergey Plis. "Explicit Group Sparse Projection with Applications to Deep Learning and NMF." _Transactions on Machine Learning Research_.
4. Yang, Huanrui, Wei Wen, and Hai Li. "Deephoyer: Learning sparser neural network with differentiable scale-invariant sparsity measures." _arXiv preprint arXiv:1908.09979_ (2019).
5. Hoyer, P. O. (2004). Non-negative matrix factorization with sparseness constraints. Journal of Machine Learning Research, 5, 1457– 1469.

**Questions:**

1. Could the authors clarify the theoretical or empirical advantages of their proposed iterative hard-thresholding heuristic (Algorithm 2) over the existing, provably-exact soft-thresholding $O(n)$ algorithm from [1, 2]?
1.1 Specifically, is this heuristic guaranteed to converge to the same optimal solution?
2. This is not an experimental benchmark paper and the paper introduces three new algorithms (Algorithms 1, 2, and 3) as mathematical optimization methods. Could the authors provide the corresponding formal guarantees for these algorithms, such as proofs of convergence, solution uniqueness, and global optimality?

---

> ### Author Response · Authors · 2025-11-16
> **A closed-form projection onto the Lorentz hypercone (original contribution).**
>
> The authors thank the reviewer for his constructive comments.
>
> Questions:
>
>    1) Could the authors clarify the theoretical or empirical advantages of their proposed iterative hard-thresholding heuristic (Algorithm 2) over the existing, provably-exact soft-thresholding
>
> Algorithm from [1, 2]? 1.1 Specifically, is this heuristic guaranteed to converge to the same optimal solution?
> This is not an experimental benchmark paper and the paper introduces three new algorithms (Algorithms 1, 2, and 3) as mathematical optimization methods. Could the authors provide the corresponding formal guarantees for these algorithms, such as proofs of convergence, solution uniqueness, and global optimality?
>
>
> **************
> Responses
>
>
> -  We provide a new convex Cone Alignement Index (CAI) (referred as extended Hoyer score).
>
>
>  - We emphasize that this GSP constraint is mathematically different from our Ratio score resulting in an iterative algorithm
>
> - Algorithm2:. A closed-form projection onto the Lorentz hypercone (original contribution).}
> To the best of our knowledge, the algorithm we introduce a closed-form projection onto the Lorentz hypercone using a single interpolation step—is entirely new.
> -We derive analytical expressions both for:\\
> - the threshold $\alpha$ that determines the active set, \\
>
> - the interpolation coefficient $\lambda$ that performs the projection.\\
>
> - The resulting closed-form solution  proceeds as follows:
> i) Ensure all components of $y$ are nonnegative.  ii) Identify the active set using the closed-form threshold $\alpha$.  iii) Compute the projection using a single interpolation with the closed-form $\lambda$.
>
> - Because the projection is performed in one step with closed-form expressions, the method does not suffer from convergence issues commonly encountered in iterative algorithms.
>
> - We provide convergence guarantees for the iterative algorithm  computing the active set through a provably correct threshold rule.
>
> - We compare our bilevel projection with the accelerated  A-HALS algorithm .  Our bilevel Closed-Form Projection (CFP) algorithm is 2r times faster than the HALS algorithm
>
> - Scope of applications.
> Our initial submission showed applications across a broad range of domains (autoencoders, single-cell analysis, Otto product classification, etc.).
> Following the reviewers’ comments, we agree to \textbf{focus the experimentation section on NLP}.
> The proposed projection operates equally well a  NLP dataset
>
> - If granted one additional page, we will include results on a complete and standardized NLP benchmark suite.
>
>
>
> We have already referenced \\
> Yang, Huanrui, Wei Wen, and Hai Li. "Deephoyer: Learning sparser neural network with differentiable scale-invariant sparsity measures." arXiv preprint arXiv:1908.09979 (2019).\\
> Hoyer, P. O. (2004). Non-negative matrix factorization with sparseness constraints. Journal of Machine Learning Research, 5, 1457– 1469.
>
> And we agree  to add the suggested references \\
>
> Ohib, Riyasat, Nicolas Gillis, Niccolo Dalmasso, Sameena Shah, Vamsi K. Potluru, and Sergey Plis. "Explicit Group Sparse Projection with Applications to Deep Learning and NMF." ICLR. 2022
>
> Thom, Markus, Matthias Rapp, and Günther Palm. "Efficient dictionary learning with sparseness-enforcing projections." International Journal of Computer Vision 114, no. 2 (2015): 168-194.

---

### Official Review · Reviewer_bMyb · 2025-10-31

**Soundness:** 3
**Presentation:** 2
**Contribution:** 2
**Rating:** 2
**Confidence:** 2

**Summary:**

This paper proposes a fast and scalable projection method for inducing structured sparsity in neural networks. It introduces an extended Hoyer score that transforms the projection space into a hypercone, allowing efficient computation. Experiments on synthetic data, autoencoders, and transformer models show faster convergence and better accuracy–sparsity trade-offs.

**Strengths:**

The paper introduces an extended Hoyer score that turns the projection space into a hypercone, enabling an efficient one-shot linear-time projection. It further extends to structured sparsity via a novel bi-level projection.

**Weaknesses:**

1. Only a few datasets (HIF2, Otto, SST-2) are tested; results could be strengthened by larger and more diverse benchmarks (e.g., ImageNet-scale, vision transformers).
2. Training setup lacks details (e.g., number of epochs, batch size, sparsity target l, and hyperparameter tuning).
3. The paper focuses on geometric intuition but lacks convergence guarantees or formal proof.
4. The paper does not compare with other sparse training methods.
5. The paper does not present computational advantage, like speed.

**Questions:**

1. Can the authors provide wall-clock runtime comparisons or FLOPs analysis on real GPUs to support the claimed linear complexity improvement?
2. Since the bilevel projection involves nested optimization, how stable is the gradient flow during end-to-end training? Any divergence observed?

---

> ### Author Response · Authors · 2025-11-17
> **wall-clock runtime comparisons or FLOPs analysis in the new version**
>
> The authors thank the reviewer for his constructive comments.
>
> Question: wall-clock runtime comparisons or FLOPs analys
>
> - Response We report in figure 2 flops since this metric is independent of the processor. while time highly depends on the processor architecture. Based on this metric, experiments shows that the Closed-form fast algorithm is approximately $6.5 $ times faster than the original Hoyer projection (depending on the data distribution). We provide a PyTorch code.
> Obviously, we get the same curves using different processor (NVIDIA GPU, Apple M3).
>
> - We provide in figure 3 the difference between our convex  solution and the non-convex Hoyer solution
>
>
> Question Only a few datasets (HIF2, Otto, SST-2) are tested; results could be strengthened by larger and more diverse benchmarks (e.g., ImageNet-scale, vision transformers).
>
> - Response
> Scope of applications.
> Our initial submission showed applications across a broad range of domains (autoencoders, single-cell analysis, Otto product classification, etc.).
> Following the reviewers’ comments, we agree to \textbf{focus the experimentation section on NLP}.
> The proposed projection operates equally well;
> If granted one additional page, we will include results on a complete and standardized NLP benchmark suite.
>
>
>
>    Question:  Training setup lacks details (e.g., number of epochs, batch size, sparsity target l, and hyperparameter tuning).
>
> - Response: We provide this details in the new version
>
>    Question:  The paper focuses on geometric intuition but lacks convergence guarantees or formal proof.
>
> - Response:We provide a closed-form projection onto the Lorentz hypercone with convergence guarantees
>
>
> Question    The paper does not compare with other sparse training methods.
>
> - Response : We Compare with Big bird. Which is considered as the best reference for sparsity
>
>  Question:   The paper does not present computational advantage, like speed.
>
> - Response : Yes, we provide  FLOPs analysis in the new version
>
> - We additionally derive a bilevel projection scheme based entirely on closed-form updates.
> Each projection admits an exact analytical solution, which eliminates numerical instabilities or convergence difficulties.
>
>
>
> -  We provide a A new Cone Alignement Index (CAI) (referred as extended Hoyer score).
>
>
> - Novelty 2. A closed-form projection (CFP) onto the Lorentz hypercone (original contribution).}
> To the best of our knowledge, the algorithm we introduce a closed-form projection onto the Lorentz hypercone using a single interpolation step—is entirely new.
> We derive analytical expressions both for:\\\\
> - the threshold $\alpha$ that determines the active set, and  \\
> We derive analytical expressions both for:\\
> - the threshold $\alpha$ that determines the active set,
>
> - the interpolation coefficient $\lambda$ that performs the projection.
>
>
> The resulting one-shot method proceeds as follows:
> i) Ensure all components of $y$ are nonnegative.
>     ii) Identify the active set using the closed-form threshold $\alpha$.
>     iii) Compute the projection using a single interpolation with the closed-form $\lambda$.

---

### Official Review · Reviewer_GnD1 · 2025-11-01

**Soundness:** 2
**Presentation:** 2
**Contribution:** 2
**Rating:** 2
**Confidence:** 5

**Summary:**

The paper tackles the question of inducing sparsity in nonconvex optimization problems using variants of the Hoyer sparsity measure. Efficient algorithms are proposed and validated on real-world datasets and deep learning architectures.

**Strengths:**

(i) Considers the Hoyer sparsity criterion in which the l1 norm is relaxed to a sum and the resulting score is analyzed to provide an efficient algorithm.
(ii) A structured sparsity setting is also briefly considered in the main paper and an algorithm is proposed to optimize for it.
(ii) Experiments are provided to show the benefits of the proposed approach on the sparse attention matrices for transformers on the OTTO group classification dataset and GLUE.

**Weaknesses:**

(a) It is unclear how the new score that is proposed satisfies desirable properties of sparsity (*)
(b) Prior works on analyzing and proposing efficient algorithms for Hoyer sparsity are not considered (*)
(c) Experiment results do not compare to other pruning/distillation approaches in the literature.

* Comparing measures of sparsity. https://arxiv.org/abs/0811.4706
* First results  on uniqueness of SPARSE NMF https://www.eurasip.org/Proceedings/Eusipco/Eusipco2005/defevent/papers/cr1658.pdf
* Block Coordinate descent for Sparse NMF https://arxiv.org/abs/1301.3527
* Sparse Activity and Sparse Connectivity in Supervised Learning. JMLR
* Explicit Group Sparse Projection with Applications to Deep Learning and NMF  https://arxiv.org/abs/1912.03896

**Questions:**

(1) What is the stated novelty of the proposed approach over the prior literature?

---

> ### Author Response · Authors · 2025-11-17
> **Novelty A closed-form projection onto the Lorentz hypercone (original contribution).**
>
> The authors thank the reviewer for his constructive comments.
>
> Question:
> (a) It is unclear how the new score that is proposed satisfies desirable properties of sparsity
>
> Response : We provide  analytical expression for the threshold $\alpha$ that determines the active set,
> any component $y_i$ smaller than $\alpha$ will be projected to zero
> \begin{equation}
>  \boxed{
>  \begin{aligned}
>    y_i \ge
>    \alpha=n^{-1}\ell_1\left(1-\sqrt{\tfrac{l(n-H(y))}{H(y)(n-l)}}\right).
>     \end{aligned}}
> \end{equation}
>
>
>
> (b) Prior works on analyzing and proposing efficient algorithms for Hoyer sparsity are not considered
>
> Response: We add reference to Explicit Group Sparse Projection with Applications to Deep Learning and NMF https://arxiv.org/abs/1912.03896 and compare with their constraint
>
> (c)What is the stated novelty of the proposed approach over the prior literature?
>
>
> - We provide A new Cone Alignement Index (CAI) (referred as extended Hoyer score).
>
>
> - Novelty 2. A closed-form projection  (CFP)  onto the Lorentz hypercone (original contribution).}
> To the best of our knowledge, the algorithm we introduce a closed-form projection onto the Lorentz hypercone using a single interpolation step—is entirely new.
> We derive analytical expressions both for:\\\\
> - the threshold $\alpha$ that determines the active set, and  \\
> We derive analytical expressions both for:\\
> - the threshold $\alpha$ that determines the active set,
>
> - the interpolation coefficient $\lambda$ that performs the projection.
>
>
> - The resulting one-shot method proceeds as follows:
> i) Ensure all components of $y$ are nonnegative.
>     ii) Identify the active set using the closed-form threshold $\alpha$.
>     iii) Compute the projection using a single interpolation with the closed-form $\lambda$.
>
> - Scope of applications.
> Our initial submission showed applications across a broad range of domains (autoencoders, single-cell analysis, Otto product classification, etc.).
> Following the reviewers’ comments, we agree to focus the experimentation section on NLP
> The proposed projection operates well on a Glue dataset
>
> - If granted one additional page, we will include results on a complete and standardized NLP benchmark suite.

---

### Author Response · Authors · 2025-11-19
**Summary comments to the  Chairman**

We sincerely thank all reviewers for their constructive evaluation and helpful suggestions. We carefully considered every comment and substantially revised the manuscript to improve clarity, rigor, and positioning with respect to prior work. Below, we address each concern with new theoretical explanations, extended experiments, and additional proofs.

- Novelty
-Reviewer comment : The proposed projection appears similar to existing sparsity projections (Hoyer, group-Hoyer, GSP 2022.)

- We agree that many works address unstructured or grouped sparsity projections. However, none of these methods solve the structured multi-vector projection onto Lorentz-type hypercones with a fully analytical closed-form projection (CFP), which is the central contribution of this work. Specifically:

- Our CAI (Cone Alignment Index) induces a Lorentz hypercone feasible space—a geometric structure that does not appear in prior sparsity formulations.

-  Our  Closed-Form Projection (CFP) provides a provably optimal projection onto this cone

- We derive analytical expressions for:
(i) computing the active set through a provably correct threshold rule with convergence guarantees.
(ii) performing  a closed-form Projection using a closed-form interpolation coefficient.

- The combination of: (i) Lorentz-based convex constraint, (ii) closed-form projection, and (iii) structured bilevel sparsification is novel and mathematically non-trivial.

- Theoretical guarantees and convergence

- Reviewer comment: Request for additional justification of theoretical foundations and convergence.

- Response: In the revised manuscript, We provide a formal theorem showing that the active‐set refinement converges to a stable fixed point.

- Reviewer comment:}CAI may simply be a re-parametrization of the Hoyer measure.

- Response: CAI is fundamentally different from Hoyer-type ratios. The key distinction: Hoyer score is a non-convex annular feasible set while CAI constraint is a convex {Lorentz hypercone. The revised manuscript now includes Figure 3 as a geometric illustration of this contrast.


-  As suggested by reviewers, We report flops since this metric is independent of the processor. while time highly depends on the processor architecture. Based on this metric, we  show that

- our Closed-form fast algorithm is approximately $6.5 $ times faster than the original Hoyer projection.

- Our bilevel Closed-Form Projection (CFP) algorithm is 2r times faster than the HALS algorithm on matrices.

- We provide code in supplementary material

- Reviewer concern: Several datasets previously used (e.g., the 779-sample HIF2 dataset, the OTTO tabular dataset) were considered too small to provide a representative evaluation of transformer sparsification methods.

- Our response: We now focus our experimental section on transformers. We removed the HIF2 and OTTO datasets.
We now provide results on:  large-scale biomedical  ECG dataset, and the standard GLUE benchmark suite for NLP.

- If one additional page is granted, we will include full experimental results on the complete GLUE leaderboard for different pretrained models (BERT-base, RoBERTa-)

- We believe the revised version now clearly demonstrates: i) a mathematically novel projection operator, ii) a closed-form, scalable, and structured sparsity control, iii)  reinforced theoretical clarity and convergence arguments , iV) a  broader and more rigorous experimental evaluation, v)  improved positioning within the context of prior literature.

- We believe that these substantial revisions directly address the reviewers’ concerns, and we respectfully request that the manuscript be reconsidered.

---

### Note · Authors · 2026-01-05

**Comment:**

Based on the Huge collusion reported, enabled deanonymization of all papers and reviews,
 We agree that it significantly increases the possibility of collusion attempt
Furthermore, while ICLR is renowned as a highly mathematical conference, we strangely received reviews of very low mathematical quality. This increases our suspicion of collusion in the initial reviews?
 We do not understand how new area chairs can review such a large number of papers.
Thus, we withdrew our paper.

**Withdrawal Confirmation:**

I have read and agree with the venue's withdrawal policy on behalf of myself and my co-authors.